# Identification of Transporter Polymorphisms Influencing Metformin Pharmacokinetics in Healthy Volunteers

**DOI:** 10.3390/jpm13030489

**Published:** 2023-03-08

**Authors:** Miriam Saiz-Rodríguez, Dolores Ochoa, Pablo Zubiaur, Marcos Navares-Gómez, Manuel Román, Paola Camargo-Mamani, Sergio Luquero-Bueno, Gonzalo Villapalos-García, Raquel Alcaraz, Gina Mejía-Abril, Estefanía Santos-Mazo, Francisco Abad-Santos

**Affiliations:** 1Research Unit, Fundación Burgos por la Investigación de la Salud (FBIS), Hospital Universitario de Burgos, 09006 Burgos, Spain; ralcaraz@hubu.es; 2Department of Health Sciences, University of Burgos, 09001 Burgos, Spain; 3Clinical Pharmacology Department, Hospital Universitario de La Princesa, Instituto Teófilo Hernando, Instituto de Investigación Sanitaria La Princesa (IP), Universidad Autónoma de Madrid (UAM), 28006 Madrid, Spain; pablo.zubiaur@salud.madrid.org (P.Z.); marcos.navares@salud.madrid.org (M.N.-G.); manuel.roman@salud.madrid.org (M.R.); paolaagueda.camargo@salud.madrid.org (P.C.-M.); sergio.luquero@salud.madrid.org (S.L.-B.); gonzalo.villapalos@salud.madrid.org (G.V.-G.); ginapaola.mejia@salud.madrid.org (G.M.-A.); francisco.abad@salud.madrid.org (F.A.-S.); 4Endocrinology Department, Hospital Universitario de Burgos, 09006 Burgos, Spain; rsantosm@saludcastillayleon.es; 5Centro de Investigación Biomédica en Red de Enfermedades Hepáticas y Digestivas (CIBERehd), Instituto de Salud Carlos III, 28029 Madrid, Spain

**Keywords:** metformin, pharmacogenetics, transporters, SLC, ABC, sex, ethnicity

## Abstract

For patients with type 2 diabetes, metformin is the most often recommended drug. However, there are substantial individual differences in the pharmacological response to metformin. To investigate the effect of transporter polymorphisms on metformin pharmacokinetics in an environment free of confounding variables, we conducted our study on healthy participants. This is the first investigation to consider demographic characteristics alongside all transporters involved in metformin distribution. Pharmacokinetic parameters of metformin were found to be affected by age, sex, ethnicity, and several polymorphisms. Age and *SLC22A4* and *SLC47A2* polymorphisms affected the area under the concentration-time curve (AUC). However, after adjusting for dose-to-weight ratio (dW), sex, age, and ethnicity, along with *SLC22A3* and *SLC22A4*, influenced AUC. The maximum concentration was affected by age and *SLC22A1*, but after adjusting for dW, it was affected by sex, age, ethnicity, *ABCG2*, and *SLC22A4*. The time to reach the maximum concentration was influenced by sex, like half-life, which was also affected by *SLC22A3*. The volume of distribution and clearance was affected by sex, age, ethnicity and *SLC22A3*. Alternatively, the pharmacokinetics of metformin was unaffected by polymorphisms in *ABCB1*, *SLC2A2*, *SLC22A2*, or *SLC47A1*. Therefore, our study demonstrates that a multifactorial approach to all patient characteristics is necessary for better individualization.

## 1. Introduction

Metformin is a hypoglycemic drug of the biguanide type used in the treatment of insulin-resistant type 2 diabetes mellitus, specially indicated for overweight patients in whom the prescribed diet and exercise are not sufficient for glycemic control [1]. Metformin is one of the first-line drugs to treat DM2 once it is diagnosed. In fact, metformin treatment is well tolerated, with mild gastrointestinal adverse events in about 30% of patients [2]. Only 5% of patients discontinue treatment due to severe intolerance [3]. Furthermore, metformin is one of the few oral DM2 drugs that leads to weight loss rather than weight gain [4].

However, the pharmacokinetics and response to metformin show great interindividual variability [5], largely dependent on the membrane transporters that regulate its transport and secretion. Metformin is not metabolized, is excreted unchanged in the urine and is distributed via the organic cation transporters OCT1, OCT2, OCT3, and OCTN1 (encoded by the genes *SLC22A1*, *SLC22A2*, *SLC22A3,* and *SLC22A4*, respectively) [6], the multidrug and toxin extrusion proteins MATE1 and MATE2 (encoded by SLC47A1 and *SLC47A2*) and the monoamine membrane transporter hENT4 (also known as PMAT), encoded by *SLC29A4*.

OCT1 is the main transporter responsible for the absorption of metformin [7]. The OCT2 transporter regulates metformin secretion in the kidney and is responsible for 80% of its clearance [8]. Some polymorphisms in *SLC22A1* were associated with reduced metformin uptake, increased elimination as a result of reduced renal tubular reabsorption and reduced therapeutic response due to decreased action of metformin in the liver [9]. Moreover, several polymorphisms in *SLC22A1* and *SLC22A2* were related to changes in pharmacokinetics and response to metformin [6,7,10,11]. Specifically, a study of 105 patients showed that metformin concentrations and changes in HbA1c at six months of treatment were lower in patients carrying decreased function alleles of *SLC22A1* rs12208357, rs72552763, and rs34059508 polymorphisms [12].

Furthermore, *SLC22A3* is expressed in multiple tissues such as the liver, kidneys, heart, skeletal muscle, and brain. This transporter appears to be relevant in the uptake of metformin in muscles [13]. Likewise, *SLC22A4* modulates metformin gastrointestinal absorption [14].

MATE1 and MATE2 transporters regulate metformin secretion in the kidney. Specifically, the *SLC47A1* rs2289669 polymorphism was associated with an increased glucose-lowering effect [11]. Although this association was proven in vitro [13], the influence of MATE1 and MATE2 on metformin disposition has not been demonstrated in diabetic patients [15], probably because of the small sample size of the study of only 48 patients. The renal and intestinal uptake of metformin is regulated by hENT4 [16]; however, there are no in vivo data demonstrating the role of this transporter in the pharmacokinetics or effects of metformin.

Likewise, the rs8192675 variant of the *SLC2A2* gene, which codes for the GLUT2 glucose transporter, was associated with a better response to metformin monotherapy (measured as glucose reduction) during the first year after diagnosis of DM2 [17,18].

Finally, P-glycoprotein (encoded by *ABCB1*) and the breast cancer resistance protein BCRP (encoded by *ABCG2*) seem to be involved in the transport of metformin in the apical membrane of the placenta [19]; and may interfere with transport in other tissues, since they are expressed in multiple locations.

Therefore, given the high interindividual variability in the pharmacokinetics and response to metformin, the study of the polymorphisms present in the main transporters that regulate its distribution could be useful to establish which patients will achieve a good response to metformin. Our aim is to identify the polymorphisms located in genes coding for transporters involved in metformin distribution that may affect its pharmacokinetics.

## 2. Materials and Methods

### 2.1. Study Population

The study population comprised 176 healthy volunteers from eight bioequivalence clinical trials performed in the Clinical Trials Unit of Hospital Universitario de La Princesa (Madrid, Spain). The protocols complied with the Spanish Legislation in clinical research in humans and were approved by the Independent Ethics Committee on Clinical Research of Hospital Universitario de La Princesa (EUDRA-CT: 2019-001393-29; EUDRA-CT: 2017-005145-79; EUDRA-CT: 2018-000401-23; EUDRA-CT: 2017-004727-73; EUDRA-CT: 2019-003274-79; EUDRA-CT: 2020-003049-12; EUDRA-CT: 2020-003619-81; EUDRA-CT: 2020-004728-40). All clinical trials were authorized by the Spanish Drug Agency and performed according to the guidelines of Good Clinical Practice. All subjects provided their written informed consent for the clinical trial and the pharmacogenetic study.

The inclusion criteria were: male and female volunteers aged from 18 to 55, free from organic or psychiatric conditions. The exclusion criteria were: history of kidney and/or liver damage, drug intake 48 h before receiving the study medication, having body mass index outside the 18.5–30.0 kg/m^2^ range, history of sensitivity to any drug and positive drug screening, smoker and daily alcohol consumer, blood donation, and pregnant or breastfeeding women.

For ethnic group comparisons, we have used the PharmGKB recommendations, which use a system of nine biogeographical groups to annotate racial and ethnic information about participants in pharmacogenomic studies [20]. The groups are based on the geographical distribution of common genetic ancestry. According to these, we have identified and categorized the volunteers into three groups: European, Latino, and African American.

### 2.2. Study Design and Procedures

We used data from eight bioequivalence clinical studies of two formulations of metformin 850–1000 mg tablets administered orally in a single dose to fasting healthy volunteers. The clinical trials were randomized, open-label, single-dose, single-center studies with a crossover design, two periods, and two sequences separated by a 28-day washout period, and metformin plasma concentrations were assessed blindly. Although the bioequivalence studies included combinations with sitagliptin or vildagliptin, only metformin data were analyzed. For pharmacokinetic analysis, 17–18 blood samples were obtained between pre-dose and 24 h post-dose. Samples were centrifuged at 4 °C for 10 min. at 3500 rpm (1900× *g*) and stored at −20 °C ± 5 °C until its shipment to an accredited external analytical laboratory. Metformin plasma concentrations were determined using high-performance liquid chromatography coupled to a tandem mass spectrometry detector (HPLC-MS/MS), with a lower limit of quantification of 2.0 ng/mL.

### 2.3. Pharmacokinetic Analysis

Pharmacokinetic parameters were calculated by a non-compartmental method using WinNonlin Professional Edition, version 7.0 or higher (Pharsight Corporation, Wilmington, DE, USA). The maximum plasma concentration (C_max_) and the time to reach the C_max_ (T_max_) were obtained directly from raw data. The area under the curve (AUC) was calculated from administration to the last measured concentration (AUC_0-t_) by linear trapezoidal integration. The total AUC from administration to infinity (AUC_∞_) was calculated as the sum of AUC_0-t_ and the residual area (Ct divided by ke, with Ct as the last measured concentration and ke as the apparent terminal elimination rate constant, which was estimated by log-linear regression from the terminal portion of concentration-time plots). Half-life (T_1/2_) was calculated by dividing 0.693 by ke. The total clearance of the drug adjusted for bioavailability (Cl/F) was calculated by dividing the dose by AUC_∞_ and adjusting for weight. The volume of distribution adjusted for bioavailability (Vd/F) was calculated as Cl/F divided by ke. AUC and Cmax were adjusted for dose and weight (AUC/dW and C_max_/dW, divided by dose/weight ratio) and logarithmically transformed for statistical analysis. For each parameter, only the reference formulation (Eucreas^®^ 50 mg/850 mg and Eucreas ^®^ 50 mg/1000 mg; Janumet ^®^ 50 mg/850 mg and Janumet ^®^ 50 mg/1000 mg; Efficib ^®^ 50 mg/1000 mg) was analyzed for each individual.

### 2.4. Safety Analysis

Throughout the study, volunteers were asked about any experienced adverse event (AE). Additionally, those AEs that were spontaneously notified by the volunteers were documented. Causality was determined using the Spanish Pharmacovigilance System algorithm [21], according to five types of AE: definite, probable, possible, unlikely and unrelated. Only definite, probable or possible AEs were considered adverse drug reactions (ADRs) and included in the statistical analysis.

### 2.5. Genotyping

DNA was extracted from 1 mL of peripheral blood samples using a DNA automatic extractor (MagNa Pure^®^ System, Roche Applied Science, Indianapolis, IN, USA) and quantified spectrophotometrically in NanoDrop^®^ ND-1000 (Wilmington, Delaware). The 260/280 absorbance ratio was used to measure the purity of the samples.

The genotyping of *ABCG2* (rs2231142), *SLC2A2* (rs8192675), *SLC22A2* (rs316019), *SLC22A3* (rs3088442), *SLC22A4* (rs272893, rs1050152), *SLC29A4* (rs3889348), *SLC47A1* (rs2289669), and *SLC47A2* (rs12943590) was outsourced and performed by MALDI-TOF mass spectrometry, with the MassARRAY^®^ platform (Agena Bioscience Inc., San Diego, CA, USA), at CEGEN-PRB3-ISCIII. All assays were performed with internal quality control, with a genotyping success rate and reproducibility of 100%. The genotyping of *ABCB1* (rs1128503, rs1045642 and rs2032582) and *SLC22A1* (rs72552763, rs12208357, and rs34059508) was performed at the Research Unit of Hospital Universitario de Burgos using TaqMan^®^ probes in a ViiA7 qPCR instrument (ThermoFisher, Waltham, MA, USA). The genes and polymorphisms were chosen for their implication in metformin transport and the functional consequences of the polymorphisms based on previous literature and minor allele frequencies.

### 2.6. Statistical Analysis

Statistical analysis was performed using SPSS 20.0 software (SPSS Inc., Chicago, IL, USA). Statistical significance was set at *p*-values lower than 0.05. The Hardy–Weinberg equilibrium was estimated for all analyzed variants. Balance deviations were detected by comparing the frequencies observed and expected using a Fisher exact test based on the De Finetti program [22].

Differences in genotype frequencies according to sex were determined using a corrected Pearson chi-square test. Differences in pharmacokinetic parameters between individuals with different sex and genotypes were analyzed by univariate parametric analysis (*t*-test or ANOVA). Multiple regression models were used to study factors related to all the pharmacokinetic dependent variables, along with ADRs. For this purpose, variables with more than two categories, such as polymorphisms, were analyzed using dummy variables. Semipartial correlations were used to determine the percentage of variance explained by each of the independent variables in the linear regression model. As this study has an exploratory observational design, we did not adjust *p*-values for multiple comparisons, according to which some authors recommend [23,24,25].

## 3. Results

### 3.1. Demographic and Genotypic Characteristics

One-hundred seventy-six healthy volunteers (91 men and 85 women) were included in the study. The mean age was 27.9 ± 7.7 years for men and 29.6 ± 8.8 years for women. Men were taller than women (1.76 ± 0.08 m vs. 1.62 ± 0.06 m, *p* < 0.001), weighed more (76.8 ± 11.0 kg vs. 61.8 ± 8.4 kg, *p* < 0.001) and exhibited a significantly higher BMI (24.7 ± 2.5 kg/m^2^ vs. 23.5 ± 3.0 kg/m^2^ for women, *p* = 0.003). Eighty-five subjects self-identified as Europeans, 89 were Latino, and two were African American.

All the genetic variants were in Hardy–Weinberg equilibrium (*p* > 0.05), except for *ABCG2* rs2231142 and *SLC22A1* rs72552763, rs12208357, and rs34059508. Genotypic frequencies are displayed according to sex and ethnicity in Table 1. The genotyping call rate was 98–100% for all analyzed polymorphisms.

No differences in genotype distribution were observed between men and women, except for *SLC47A2* rs12943590. Moreover, some differences were observed related to *ABCG2*, *SLC2A2*, *SCL22A1*, *SLC22A3,* and *SCL22A4* genotype distribution among different ethnic groups (1). The percentage of subjects carrying *ABCG2* rs2231142 G/G genotype was lower among Latinos compared to other ethnic groups. The two African American subjects were carriers of the *SLC2A2* rs8192675 C/C genotype and *SLC22A1* rs72552763 GAT/GAT genotype, in contrast to the much lower percentage of carriers of both polymorphisms among other ethnic groups. Indeed, no African American subject carried the *SLC22A3* rs3088442 A/A genotype, while both African American subjects carried the *SLC22A4* rs1050152 C/C genotype, being these differences statistically significant.

### 3.2. Pharmacokinetic Analysis

The mean and standard deviation (SD) of pharmacokinetic parameters is shown in Table 2. Metformin pharmacokinetic parameters were affected by sex, as women showed a significantly higher Cmax that was also significant but, on the contrary, lower when corrected by dose and weight (Cmax/dW). Likewise, women showed a significantly lower Tmax but a higher T1/2. Moreover, when stratifying individuals by ethnicity, we observed a significantly higher AUC/dW in African American individuals (*p* < 0.05).

The univariate analysis revealed an association between some pharmacokinetic parameters and polymorphisms in *ABCB1* (rs1128503, rs2032582), *ABCG2* (rs2231142), *SLC2A2* (rs8192675), *SLC22A3* (rs3088442) and *SLC22A4* (rs1050152, rs272893) (Table 3). However, in the multivariate analysis, only *ABCG2* (rs22311442), *SLC22A1* (rs72552763), *SLC22A3* (rs3088442), *SLC22A4* (rs1050152, rs272893), and *SLC47A2* (rs12943590) remained statistically significant (Table 4).

In the multivariate analysis, sex and ethnicity continue to influence some of the pharmacokinetic parameters (Table 4). Age and SLC22A4 and SLC47A2 polymorphisms affected AUC. However, after adjusting for dose-to-weight ratio (dW), sex, age, and ethnicity, along with SLC22A3 and SLC22A4, influenced AUC. C_max_ was affected by age and SLC22A1, but after adjusting for dW, it was affected by sex, age, ethnicity, ABCG2, and SLC22A4. T*max* was influenced by sex, like T1/2, which was also affected by SLC22A3. Vd/F and Cl/F were affected by sex, age, ethnicity and SLC22A3.

### 3.3. Safety Profile

In all, 20 subjects experienced an ADR, which is 11.36% of the 176 total. Ten volunteers had a headache (5.7%), five had diarrhea (2.8%), three had nausea (1.7%), and one each had hypoglycemia, stomachache, and dizziness (0.6% each). In the logistic regression model, being a woman appeared to be a risk factor for developing ADR (OR: 11.5, 95% CI: 2.4–54.8), as well as being a heterozygote (OR: 7.9, 95% CI: 1.5–41.6) and homozygote (OR: 9.2, 95% CI: 1.3–62.8) carrier of *ABCB1* rs2032582. ADR occurrence was unaffected by any other polymorphism.

## 4. Discussion

For patients with DM2, metformin is the most often recommended medication. Metformin is used by more than 120 million type 2 diabetes patients globally [26]. However, individual differences in the pharmacological response to metformin are significant. In 35% of patients, metformin did not achieve the best glycaemic control, which called for either a dose increase or the use of a combination of hypoglycaemic drugs [27]. Some interindividual variations in the glycaemic response to metformin are influenced by genetics. Numerous pharmacogenetic investigations have shown that differences in the genes that encode the transporters responsible for the pharmacokinetics and pharmacodynamics of metformin are often related to metformin response [26]. Metformin’s total exposure and efficacy can be increased or decreased by certain genetic polymorphisms; hence, these variations must be examined to get the optimal therapeutic outcome.

Our study in healthy volunteers provides a good environment to study the individual influence of transporter polymorphisms on metformin pharmacokinetics without confounding factors. However, it is not the real environment of metformin therapy, where patients usually present comorbidities and concomitant treatment that can influence the response to metformin. Therefore, future perspectives should include studies exploring the relationship between metformin and its metabolites and type 2 diabetes risk and symptoms, which may shed light on the disease’s pathophysiological features and lead to the identification of novel, reliable biomarkers that may aid in the early detection and treatment of type 2 diabetes [26].

SLC22A1 decreased function variants were related to a diminished response to metformin in some patients [28]. Following an oral glucose tolerance test, *SLC22A1* rs72552763, *SLC22A1* rs12208357, and *SLC22A1* rs34059508 were associated with a reduced capacity of metformin to lower blood sugar in a study performed on 20 healthy human volunteers [10]. In our study, which had a much higher sample size, only *SLC22A1* rs72552763 seemed to affect the pharmacokinetic parameters of metformin. However, there were some conflicting findings among patients. *SLC22A1* rs72552763, rs12208357, and rs34059508 decreased-function variants were similarly linked with lower trough concentrations of metformin and a reduction in HbA1c following six months of metformin therapy in a sample of 151 type 2 diabetes patients [12]. In addition, it has been postulated that decreased metformin transport through *SLC22A1* might raise metformin levels in the intestine, hence increasing the likelihood of severe gastrointestinal adverse effects and drug withdrawal [29]. Individuals with two *SLC22A1* reduced-function alleles who were administered *SLC22A1* inhibitors were nearly four times more likely to exhibit intolerance [29]. Inadequate SLC22A1 transport appears to be a major contributor to the metformin safety profile, as suggested by these results. Conversely, research conducted on 33 individuals indicated that polymorphisms in *SLC22A1* and *SLC22A2* contributed little to the clinical effectiveness of metformin [30]. Likewise, *SLC22A1* rs72552763 and *SLC22A1* rs12208357 polymorphisms were not associated with glycaemic response to metformin in research involving 1531 patients with type 2 diabetes [31]. Some *SLC22A1* variants have an impact on metformin responsiveness; nevertheless, the limited number of patients and the co-medication of other anti-hyperglycaemic medicines in some research groups represent a disadvantage and explain the inconsistencies between the mentioned studies. To properly explore the combined effect of *SLC22A1* function on the pharmacokinetic, pharmacodynamic, and safety profiles of metformin, it is necessary to conduct additional research using a large patient cohort sample with no other anti-hyperglycaemic concomitant therapy.

The study by Abrahams-October et al. describes an association between *SLC22A2* rs316009 and improved response to metformin treatment in 140 indigenous sub-Saharan African patients [32]. No association was found between *SLC22A1* and *SLC2A2* [32]. Nevertheless, as already mentioned, a previous study found that *SLC22A2* polymorphisms had a negligible influence on metformin’s clinical efficacy [30]. Moreover, there was no correlation between *SLC22A2* variants and metformin response, according to a growing body of research [28,33,34,35]. Our results are consistent with those previously described since we found no differences between the different *SLC22A2* rs316019 genotypes and pharmacokinetic parameters.

On the other hand, it has been shown that the C allele of the variant rs8192675 in the gene *SLC2A2*, which codes for the glucose transporter (GLUT2), is essential for regulating the activity of metformin [18]. Our results indicated that those with the C/C genotype for *SLC2A2* rs8192675 had greater values for Vd/F and Cl/F than those with the T allele. Since it plays no role in the pharmacokinetics of metformin and is absent from the multivariate analysis, it is possible that this discovery is merely coincidental.

Regarding OCT3, we found lower AUC/dW in individuals carrying the rs3088442 A/A genotype. A previous study by Tzvetkov et al. [9] described no significant associations between the renal clearance of metformin and variants analyzed in *SLC22A2*, *SLC22A3*, *SLC22A4*, and *SLC47A1*. In our study, carriers of *SLC22A4* rs1050152 and rs272893 mutated genotypes showed lower Cmax/dW and T1/2, respectively. Moreover, the multivariate analysis revealed that *SLC22A3* and *SLC22A4* polymorphisms affected AUC, AUC/dW, C_max_/dW, T_1/2_, Vd/F and Cl/F; likewise, *SLC47A2* rs1294359 influenced AUC. The study by Chen et al. revealed that several *SLC22A3* variants showed altered substrate specificity and, therefore, polymorphisms affecting OCT3 function may modulate metformin action [13]. Regarding OCTN1, our results contradict previous studies. In a metanalysis performed including 7968 individuals, no influence was found of the *SLC22A4* analyzed variants [35]. This metanalysis also evaluated other polymorphisms in *SLC22A1*, *SLC22A2*, *SLC47A1*, and *SLC47A2* without finding significant results [35]. We were unable to find differences in any pharmacokinetic parameters among different *SLC47A1* rs2289669 genotypes, consistent with the previously reported by Kalamajski et al., who describe that none of the most commonly analyzed variants in *SLC47A1*, including rs2289669, showed significant association with metformin response [36]. In light of the aforementioned and the metanalysis findings, it may be concluded that candidate variations in transporter genes may not be effective as predictors for personalized therapy with metformin, despite their importance in metformin pharmacokinetics [35]. There is a need for more studies where the impact of transporters variants on the pharmacokinetic parameters and effectiveness of metformin could be evaluated jointly.

Lastly, P-glycoprotein (encoded by *ABCB1*) and the breast cancer resistance protein BCRP (encoded by *ABCG2*) appear to be involved in metformin transport in the apical membrane of the placenta [19]. Furthermore, since they are expressed in a variety of tissues, they may also interfere with transport within these locations. Metformin transporters OCT1, OCT2, OCT3, and P-glycoprotein were linked to therapeutic failure in a recent pharmacogenetics investigation of Mexican individuals with type 2 diabetes [19]. Another study involving 103 patients with type 2 diabetes mellitus revealed no correlation between *ABCB1* polymorphism and HbA1c stabilization [37]. However, in our study, the T/T genotype of *ABCB1* rs1128503 polymorphism was related to lower Vd/F and Cl/F. Likewise, heterozygous genotypes of *ABCB1* rs2032582 correlate with higher C_max_/dW and lower Vd/F and Cl/F. However, all these associations disappear in the multivariate analysis, which is consistent with the results previously found in the literature. On the contrary, *ABCG2* rs2231142 G/G genotype was significantly associated with a lower C_max_/dW both in the univariate and multivariate analysis. The polymorphism *ABCG2* c.421C > A (rs2231142) results in reduced activity of the important drug efflux transporter [38]. According to our knowledge, ours is the first study to associate *ABCG2* polymorphism with diminished metformin concentration, which surely warrants further evaluation.

A previous study reported that African-Americans tend to have a more favorable glycemic response to metformin than European Americans [39]. Our results align with these previous findings since African American participants had a significantly greater AUC after adjusting for dosage and weight. However, due to the ethnic composition of our region, just two of the participants were African American. Additional research is required to evaluate if this leads to proportional reductions in diabetic complications in patients based on their ethnic origin. Moreover, based on our review of the available literature, our study is the first to report sex and ethnicity differences in metformin pharmacokinetic parameters related to transporters polymorphisms. The genotype distributions of *ABCG2* rs2231142, *SLC2A2* rs8192675, *SLC22A1* rs72552763, *SLC22A4* rs3088442, and *SCL22A4* rs1050152 varied among different ethnic groups. Interestingly, some of these polymorphisms have been demonstrated to also affect the pharmacokinetic parameters of metformin. Precision medicine advocates taking into account all intrinsic (e.g., age, genetics, sex, ethnicity) and extrinsic (e.g., concomitant medication, comorbidities, physical activity) patient factors. Therefore, our study demonstrates that a multifactorial approach to all patient characteristics is necessary for better individualization.

Unfortunately, the majority of recent clinical investigations were limited in scope and only included participants of a single ethnicity [40]. As a consequence, the link between genetic indicators and the combined clinical outcomes has to be investigated in a more extensive study with a greater population and more complicated clinical situations [40]. Additionally, there are studies with contradicting findings using the same criteria [40]. The discrepancy in the results can be attributed to variables including population size, ethnic background, and analysis tools [40]. Better antidiabetic therapy could be developed with a better knowledge of the pharmacogenomics of drugs now used to treat type 2 diabetes mellitus.

### Study Limitations

Our main limitation is the small proportion of African American in our population, which preclude us from finding a higher proportion of volunteers in this ethnic group. Moreover, as a candidate-gene study, there is a possibility that other variants could be related to metformin pharmacokinetics and were not included in the analysis. Another limitation is the lack of multiple comparison correction, which could have led to false positive results. However, some experts recommend not correcting for multiple testing when analyzing data [23,24,25]. Indeed, it is recommended to account for multiple comparisons once interpreting the results rather than in calculations. In fact, some authors state that the use of multiple comparison correction should be avoided to perform empirical research since there is a potential cost of many more false negatives when controlling for false positives [41,42].

## 5. Conclusions

This is the first investigation to consider intrinsic characteristics, including age, sex, and ethnicity, alongside all transporters involved in metformin distribution. Pharmacokinetic parameters of metformin were found to be affected by age, sex, ethnicity, and several polymorphisms in *ABCG2*, *SLC22A1*, *SLC22A3*, *SLC22A4*, and *SLC47A2*. Alternatively, the pharmacokinetics of metformin was unaffected by polymorphisms in *ABCB1*, *SLC2A2*, *SLC22A2*, or *SLC47A1*. Our study demonstrates that a multifactorial approach to all patient characteristics is necessary for better individualization.

## Figures and Tables

**Table 1 jpm-13-00489-t001:** Genotype frequencies of transporters in the study subjects, stratified by sex and ethnicity. Values are expressed as the number of individuals (%).

Gene	n	Men (*n* = 91)	Women (*n* = 85)	African American (*n* = 2)	European (*n* = 85)	Latino (*n* = 89)
** *ABCB1* ** **rs1045642**						
C/C	59	30 (33.0)	29 (34.1)	0 (0.0)	27 (31.8)	32 (36.0)
C/T	80	42 (46.2)	38 (44.7)	2 (100.0)	43 (50.6)	35 (39.3)
T/T	37	19 (20.9)	18 (21.2)	0 (0.0)	15 (17.6)	22 (24.7)
** *ABCB1* ** **rs1128503**						
C/C	69	37 (40.7)	32 (37.6)	2 (100.0)	33 (38.8)	34 (38.2)
C/T	81	41 (45.1)	40 (47.1)	0 (0.0)	40 (47.1)	41 (46.1)
T/T	26	13 (14.3)	13 (15.3)	0 (0.0)	12 (14.1)	14 (15.7)
** *ABCB1* ** **rs2032582**						
G/G	65	34 (37.8)	31 (36.9)	2 (100.0)	31 (36.9)	32 (36.4)
G/A + G/T	81	40 (44.4)	41 (48.8)	0 (0.0)	40 (47.6)	41 (46.6)
A/A + T/T	28	16 (17.8)	12 (14.3)	0 (0.0)	13 (15.5)	15 (17.0)
** *ABCG2* ** **rs2231142 ˜**						
G/G	134	69 (76.7)	65 (76.5)	2 (100.0)	75 (88.2)	57 (64.8)
G/T	34	18 (20.0)	16 (18.8)	0 (0.0)	9 (10.6)	25 (28.4)
T/T	7	3 (3.3)	4 (4.7)	0 (0.0)	1 (1.2)	6 (6.8)
** *SLC2A2* ** **rs8192675 ˜**						
T/T	79	39 (43.3)	40 (47.1)	0 (0.0)	37 (44.0)	42 (47.2)
T/C	79	40 (44.4)	39 (45.9)	0 (0.0)	41 (48.8)	38 (42.7)
C/C	17	11 (12.2)	6 (7.1)	2 (100.0)	6 (7.1)	9 (101)
** *SLC22A1* ** **rs72552763 ˜**						
GAT/GAT	114	65 (71.4)	49 (57.6)	2 (100.0)	63 (74.1)	49 (55.1)
GAT/delGAT	47	22 (24.2)	25 (29.4)	0 (0.0)	19 (22.4)	28 (31.5)
delGAT/delGAT	15	4 (4.4)	11 (12.9)	0 (0.0)	3 (3.5)	12 (13.5)
** *SLC22A1* ** **rs12208357**						
C/C	164	83 (91.2)	81 (95.3)	2 (100.0)	78 (91.8)	84 (94.4)
C/T	10	7 (7.7)	3 (3.5)	0 (0.0)	5 (5.9)	5 (5.6)
T/T	2	1 (1.1)	1 (1.2)	0 (0.0)	2 (2.4)	0 (0.0)
** *SLC22A1* ** **rs34059508**						
G/G	172	90 (98.9)	82 (96.5)	2 (100.0)	84 (98.8)	86 (96.6)
G/A	3	0 (0.0)	3 (3.5)	0 (0.0)	0 (0.0)	3 (3.4)
A/A	1	1 (1.1)	0 (0.0)	0 (0.0)	1 (1.2)	0 (0.0)
** *SLC22A2* ** **rs316019**						
C/C	143	78 (85.7)	65 (76.5)	1 (50.0)	66 (77.6)	76 (85.4)
C/A	30	13 (14.3)	17 (20.0)	1 (50.0)	17 (20.0)	12 (13.5)
A/A	3	0 (0.0)	3 (3.5)	0 (0.0)	2 (2.4)	1 (1.1)
** *SLC22A3* ** **rs3088442 ˜**						
G/G	81	39 (43.3)	42 (49.4)	1 (50.0)	49 (57.6)	31 (35.2)
G/A	72	38 (42.2)	34 (40.0)	1 (50.0)	29 (34.1)	42 (47.7)
A/A	22	13 (14.4)	9 (10.6)	0 (0.0)	7 (8.2)	15 (17.0)
** *SLC22A4* ** **rs1050152 ˜**						
C/C	80	39 (43.3)	41 (48.2)	2 (100.0)	25 (29.4)	53 (60.2)
C/T	77	45 (50.0)	32 (37.6)	0 (0.0)	46 (54.1)	31 (35.2)
T/T	18	6 (6.7)	12 (14.1)	0 (0.0)	14 (16.5)	4 (4.5)
** *SLC22A4* ** **rs272893**						
C/C	68	35 (38.5)	33 (39.3)	1 (50.0)	33 (38.8)	34 (38.6)
C/T	86	47 (51.6)	39 (46.4)	0 (0.0)	42 (49.4)	44 (50.0)
T/T	21	9 (9.9)	12 (14.3)	1 (50.0)	10 (11.8)	10 (11.4)
** *SLC29A4* ** **rs3889348**						
G/G	76	40 (44.4)	36 (42.9)	0 (0.0)	32 (38.1)	44 (50.0)
G/A	82	40 (44.4)	42 (50.0)	2 (100.0)	45 (53.6)	35 (39.8)
A/A	16	10 (11.1)	6 (7.1)	0 (0.0)	7 (8.3)	9 (10.2)
** *SLC47A1* ** **rs2289669**						
G/G	51	29 (32.2)	22 (25.9)	1 (50.0)	21 (24.7)	29 (33.0)
G/A	90	48 (53.3)	42 (49.4)	1 (50.0)	46 (54.1)	43 (48.9)
A/A	34	13 (14.4)	21 (24.7)	0 (0.0)	18 (21.2)	16 (18.2)
** *SLC47A2* ** **rs12943590**						
G/G	83	44 (48.4)	39 (46.4)	0 (0.0)	42 (49.4)	41 (46.6)
G/A	77	35 (38.5)	42 (50.0)	2 (100.0)	33 (38.8)	42 (47.7)
A/A	15	12 (13.2)	3 (3.6) *	0 (0.0)	10 (11.8)	5 (5.7)

Values are expressed as the number of individuals (%). * *p* < 0.05 vs. men ˜ *p* < 0.05 among ethnicity groups.

**Table 2 jpm-13-00489-t002:** Pharmacokinetic parameters of metformin after a single 850–1000 mg oral dose.

Pharmacokinetic Parameter	Total (*n*= 176)	Men (*n* = 91)	Women (*n* = 85)	*p*-Value	African American (*n* = 2)	European (*n* = 85)	Latino (*n* = 89)	*p*-Value
AUC (ng·h/mL)	13,200.6 (2895.7)	12,945.7 (2754.2)	13,473.4 (3032.5)	0.26	14,476.9 (3384.9)	13,255.6 (2373.4)	13,268.4 (3041.1)	0.77
AUC/dW (ng·h·kg/mL·mg)	966.4 (240.4)	1042.5 (224.6)	884.8 (231.1)	0.18	1368.8 (309.3)	999.4 (199.1)	936.3 (245.9)	0.01
C_max_ (ng/mL)	1509.6 (344.1)	1464.2 (357.4)	1558.2 (324.4)	0.05	1580.6 (46.1)	1473.3 (324.1)	1542.6 (364.4)	0.40
C_max_/dW (ng·kg/mL·mg)	110.3 (27.5)	117.8 (27.6)	102.3 (25.2)	<0.01	149.6 (3.2)	111.1 (26.4)	108.7 (28.3)	0.14
T_max_ (h)	4.3 (1.3)	4.5 (1.4)	4.1 (1.2)	0.02	5.5 (2.1)	4.4 (1.0)	4.2 (1.5)	0.17
T_1/2_ (h)	4.2 (0.7)	4.0 (0.6)	4.5 (0.6)	<0.01	3.8 (0.3)	4.2 (0.6)	4.3 (0.6)	0.30
Vd/F (mL/kg)	6335.5 (2951.6)	5236.0 (2191.0)	7499.7 (3208.3)	0.99	4067.6 (563.8)	5941.2 (2523.9)	6767.9 (3287.9)	0.85
Cl/F (mL/h·kg)	1009.4 (389.4)	886.8 (327.7)	1139.2 (409.0)	0.39	749.7 (169.4)	986.9 (334.5)	1054.4 (435.6)	0.70

Values are shown as mean (SD). Abbreviation: AUC, area under the curve; C_max_, maximum plasma concentration; T_max_, time to reach the maximum plasma concentration; T_1/2_, half-life; Cl/F, total drug clearance adjusted for bioavailability; Vd/F, volume of distribution adjusted for bioavailability; dW, adjusted for dose and weight ratio.

**Table 3 jpm-13-00489-t003:** Association between metformin pharmacokinetic parameters and polymorphisms in the studied transporters.

Gene	n	AUC (ng·h/mL)	AUC/dW (ng·h·kg/mL·mg)	C_max_ (ng/mL)	C_max_/dW (ng·kg/mL·mg)	T_max_ (h)	T_1/2_ (h)	Vd/F (L/kg)	Cl/F (L/h·kg)
** *ABCB1* ** **rs1045642**									
C/C	59	13,047.5 (2752.5)	962.9 (209.5)	1486.6 (349.1)	109.2 (24.5)	4.3 (1.3)	4.3 (0.7)	6644.5 (2491.0)	1048.8 (298.9)
C/T	79	13,472.5 (3128.0)	968.6 (259.4)	1560.4 (362.4)	112.3 (30.1)	4.3 (1.4)	4.2 (0.8)	6108.1 (3131.1)	985.5 (423.4)
T/T	37	12,856.6 (2600.0)	967.1 (250.2)	1436.4 (281.3)	107.7 (26.4)	4.3 (1.3)	4.3 (0.5)	6328.4 (3255.6)	997.6 (443.6)
** *ABCB1* ** **rs1128503**									
C/C	69	13,047.5 (2752.5)	962.9 (209.5)	1486.6 (349.1)	109.2 (24.5)	4.3 (1.3)	4.3 (0.7)	6554.6 (2415.6)	1039.9 (274.2)
C/T	80	13,643.0 (2748.3)	980.8 (236.6)	1560.4 (362.4)	112.3 (30.1)	4.3 (1.4)	4.2 (0.6)	6312.8 (3136.9)	1015.1 (421.6)
T/T	26	12,856.6 (2600.0)	967.1 (250.2)	1436.4 (281.3)	107.7 (26.4)	4.3 (1.3)	4.3 (0.5)	5824.0 (3641.4) *	910.6 (525.7) *
** *ABCB1* ** **rs2032582**									
G/G	65	13,290.3 (2633.0)	996.3 (222.7)	1460.3 (284.1)	109.4 (24.7)	4.4 (1.2)	4.3 (0.6)	6617.7 (2255.3)	1055.4 (243.7)
G/A + G/T	80	13,434.5 (2771.7)	958.7 (226.8)	1588.2 (384.5)	113.4 (30.2) *	4.2 (1.4)	4.2 (0.6)	5965.6 (3187.0) *	962.8 (448.6) *
A/A + T/T	28	12,921.6 (2913.6)	959.3 (258.1)	1400.1 (317.9)	103.2 (25.2)	4.3 (1.1)	4.4 (0.5)	6705.6 (3652.9)	1023.1 (478.5)
** *ABCG2* ** **rs2231142**									
G/G	133	13,121.5 (2696.9)	965.8 (225.8)	1461.3 (313.8) *	107.2 (24.8)	4.4 (1.3)	4.3 (0.6)	6373.8 (2930.0)	1017.2 (387.2)
G/T	34	13,451.9 (2341.4)	982.4 (229.2)	1639.1 (315.0)	119.6 (28.6)	4.1 (1.3)	4.2 (0.6)	6400.1 (2831.8)	1021.2 (361.3)
T/T	7	14,618.3 (2704.4)	986.4 (311.3)	1593.2 (454.0)	109.0 (38.4)	4.1 (0.9)	4.5 (0.6)	5628.6 (4255.6)	839.2 (578.5)
** *SLC2A2* ** **rs8192675**									
T/T	78	13,644.2 (2753.7)	962.6 (226.7)	1536.4 (364.1)	108.3 (27.5)	4.3 (1.6)	4.3 (0.6)	6214.4 (3217.9)	975.7 (443.4) *
T/C	79	13,083.9 (2797.0)	981.7 (227.0)	1501.6 (344.3)	112.6 (28.3)	4.3 (1.0)	4.2 (0.6)	6218.2 (2617.9)	1019.0 (341.1)
C/C	17	12,501.3 (2154.2)	970.0 (272.2)	1414.5 (237.7)	108.2 (24.8)	4.2 (1.4)	4.5 (0.7)	7417.6 (3179.3) *	1117.0 (342.7)
** *SLC22A1* ** **rs72552763**									
GAT/GAT	114	13,129.9 (2547.6)	988.3 (220.3)	1468.6 (289.3)	110.7 (26.1)	4.4 (1.3)	4.2 (0.6)	6156.5 (2817.9)	994.1 (361.9)
GAT/delGAT	46	13,605.9 (3218.2)	964.3 (253.4)	1578.6 (459.2)	111.0 (30.8)	4.1 (1.5)	4.4 (0.6)	6617.6 (2781.5)	1044.1 (382.4)
delGAT/delGAT	15	13,374.6 (2465.8)	870.1 (209.6)	1604.4 (269.7)	105.0 (28.2)	4.2 (1.1)	4.4 (0.6)	6831.1 (4304.8)	1019.3 (593.0)
** *SLC22A1* ** **rs12208357**									
C/C	163	13,328.0 (2781.2)	976.7 (232.6)	1515.1 (345.6)	110.8 (27.5)	4.3 (1.3)	4.3 (0.6)	6253.6 (2968.4)	999.4 (397.1)
C/T	10	12,540.1 (1846.5)	898.7 (119.3)	1402.8 (346.6)	100.3 (21.1)	4.7 (1.2)	4.4 (1.0)	7358.9 (2375.1)	1131.0 (155.1)
T/T	2	12,719.9 (1389.8)	942.6 (474.9)	1589.5 (217.7)	118.4 (62.5)	3.0 (0.0)	4.4 (0.4)	7894.6 (4599.9)	1215.1 (612.2)
** *SLC22A1* ** **rs34059508**									
G/G	171	13,315.8 (2734.2)	974.9 (228.2)	1511.6 (345.7)	110.4 (25.2)	4.3 (1.3)	4.3 (0.6)	6284.0 (2937.8)	1003.1 (388.9)
G/A	3	10,755.3 (821.8)	699.8 (82.3)	1286.0 (139.7)	83.3 (6.2)	4.2 (1.0)	4.8 (0.6)	10,005.7 (1205.9)	1442.3 (8171.7)
A/A	1	14,032.7	1267.9	1833.9	165.7	5.5	3.6	4126.7	788.7
** *SLC22A2* ** **rs316019**									
C/C	142	13,371.3 (2275.8)	976.7 (225.6)	1519.6 (353.9)	110.9 (27.7)	4.4 (1.3)	4.2 (0.6)	6241.5 (2999.0)	1000.5 (393.8)
C/A	30	12,845.0 (2586.4)	966.4 (256.2)	1463.4 (311.6)	109.2 (27.9)	4.1 (1.5)	4.4 (0.6)	6600.4 (2827.8)	1026.6 (385.7)
A/A	3	12,981.0 (1518.4)	798.0 (40.9)	1495.2 (152.9)	92.0 (3.5)	5.2 (0.3)	4.5 (0.7)	8137.9 (1165.3)	1255.4 (65.6)
** *SLC22A3* ** **rs3088442**									
G/G	81	13,502.3 (2613.2)	1010.8 (231.3)	1521.4 (303.1)	113.5 (26.1)	4.4 (1.1)	4.2 (0.5)	5915.4 (2704.1)	959.3 (368.1)
G/A	71	13,300.5 (2842.9)	950.4 (219.3)	1503.1 (354.8)	107.6 (29.3)	4.3 (1.5)	4.3 (0.7)	6398.4 (2915.3)	1016.0 (386.3)
A/A	22	12,128.5 (2392.7)	882.0 (226.2) *	1419.6 (312.9)	102.6 (27.0)	4.3 (1.3)	4.4 (0.6)	7785.7 (3583.1)	1183.8 (445.2)
** *SLC22A4* ** **rs1050152**									
C/C	79	13,663.5 (3004.2)	999.2 (257.6)	1554.2 (378.6)	113.5 (29.9)	4.3 (1.5)	4.3 (0.6)	6169.9 (2911.1)	984.9 (399.7)
C/T	77	13,109.3 (2535.0)	968.8 (199.8)	1497.4 (307.2)	110.6 (25.3)	4.3 (1.0)	4.2 (0.6)	6363.5 (2809.0)	1017.9 (360.2)
T/T	18	12,302.7 (2010.2)	870.5 (205.2)	1360.3 (311.8)	94.8 (21.3) *	4.5 (1.4)	4.4 (0.8)	6951.0 (3804.7)	1072.8 (480.1)
** *SLC22A4* ** **rs272893**									
C/C	68	12,821.3 (2695.6)	930.6 (232.6)	1473.4 (292.5)	106.4 (24.8)	4.4 (1.3)	4.3 (0.6)	6779.4 (3124.1)	1076.8 (399.5)
C/T	85	13,393.3 (2737.2)	993.2 (232.7)	1515.7 (370.2)	112.2 (28.9)	4.4 (1.3)	4.2 (0.6)	6132.5 (2936.3)	976.9 (389.5)
T/T	21	14,432.8 (2464.2) *	1029.5 (191.6)	1614.8 (383.5)	116.0 (29.9)	3.6 (1.5)*	4.3 (0.7)	5622.1 (2300.7)	907.7 (334.3)
** *SLC29A4* ** **rs3889348**									
G/G	76	13,401.2 (3013.1)	969.1 (230.6)	1495.0 (329.8)	107.9 (25.6)	4.2 (1.3)	4.3 (0.6)	6485.2 (3262.2)	1009.2 (430.2)
G/A	81	13,252.1 (2420.9)	973.8 (228.3)	1540.2 (322.5)	113.2 (28.3)	4.4 (1.3)	4.2 (0.6)	6079.8 (2705.1)	992.4 (373.7)
A/A	16	13,158.4 (2880.5)	996.8 (250.3)	1488.1 (471.7)	111.5 (30.8)	4.5 (1.2)	4.2 (0.7)	6696.5 (2789.1)	1065.0 (272.7)
** *SLC47A1* ** **rs2289669**									
G/G	50	13,571.3 (2656.6)	1008.3 (244.3)	1502.1 (291.0)	111.5 (26.8)	4.5 (1.5)	4.1 (0.5)	5949.7 (2822.3)	973.7 (392.4)
G/A	90	13,212.5 (2901.1)	967.8 (216.5)	1514.5 (361.0)	111.1 (27.6)	4.2 (1.3)	4.3 (0.6)	6229.3 (3086.7)	989.4 (404.6)
A/A	34	12,857.7 (2201.3)	918.9 (233.4)	1464.1 (282.3)	103.4 (22.3)	4.5 (1.0)	4.4 (0.6)	7253.0 (2668.4)	1122.2 (333.8)
** *SLC47A2* ** **rs12943590**									
G/G	82	13,652.2 (2895.6)	988.6 (249.7)	1521.4 (352.3)	109.9 (29.0)	4.3 (1.5)	4.3 (0.6)	6156.9 (3257.3)	983.7 (433.7)
G/A	77	13,043.0 (2642.7)	953.4 (215.9	1524.5 (356.5)	111.2 (27.4)	4.2 (1.3)	4.3 (0.6)	6642.9 (2742.4)	1043.4 (357.5)
A/A	15	12,637.8 (1875.3)	989.9 (185.8)	1386.3 (199.7)	108.8 (20.8)	4.8 (0.6)	4.0 (0.6)	5597.2 (2114.0)	954.1 (287.0)

Values are shown as mean (SD). Abbreviation: AUC, area under the curve; C_max_, maximum plasma concentration; T_max_, time to reach the maximum plasma concentration; T_1/2_, half-life; Cl/F, total drug clearance adjusted for bioavailability; Vd/F, volume of distribution adjusted for bioavailability; dW, adjusted for dose and weight ratio. * *p* < 0.05.

**Table 4 jpm-13-00489-t004:** Factors influencing metformin pharmacokinetic parameters. Results from the multivariate analysis.

	Pharmacokinetic Parameters							
Independent Variable	AUC (ng·h/mL)	AUC/dW (ng·h·kg/mL·mg)	C_max_ (ng/mL)	C_max_/dW (ng·kg/mL·mg)	T_max_ (h)	T_1/2_ (h)	Vd/F (L/kg)	Cl/F (L/h·kg)
Sex	---	−168.9 (−228.3, −109.5)	---	−14.6 (−21.9, −7.4)	−0.5 (−0.9, −0.1)	0.5 (0.3, 0.7)	2358.6 (1545.5, 3171.7)	274.2 (163.0, 385.4)
Age	94.6 (47.8, 414.4)	9.7 (5.9, 13.4)	12.3 (6.7, 18.0)	1.0 (0.6, 1.5)	---	---	−89.4 (−140.6, −38.1)	−9.4 (−16.0, −2.7)
Ethnicity	---	−126.9 (−190.3, −63.4)	---	−14.9 (−22.7, −7.2)	---	---	975.9 (132.3, 1819.4)	---
*ABCG2* rs2231142	---	---	---	7.4 (0.3, 14.4)	---	---	---	---
*SLC22A1* rs72552763	---	---	74.4 (2.7, 146.0)	---	---	---	---	---
*SLC22A3* rs3088442	---	−53.3 (−96.9, −9.6)	---	---	---	0.1 (0.0, 0.3)	762.7 (165.2, 1360.2)	110.9 (31.0, 190.8)
*SLC22A4* rs272893	716.8 (126.7, 1306.8)	---	---	---	---	---	---	---
*SLC22A4* rs1050152		−57.9 (−103.9, −11.9)	---	−6.2 (−11.8, −0.6)	---	---	---	---
*SLC47A2* rs12943590	−658.7 (−1280.2, −37.3)	---	---	---	---	---	---	---
R^2^	0.558	0.331	0.124	0.239	0.038	0.209	0.250	0.176

Abbreviation: Values shown as β (non-standardized β coefficient) expressed as number (CI 95%); AUC, area under the curve; C_max_, maximum plasma concentration; T_max_, time to reach the maximum plasma concentration; T_1/2_, half-life; Cl/F, total drug clearance adjusted for bioavailability; Vd/F, volume of distribution adjusted for bioavailability; dW, adjusted for dose and weight ratio.

## Data Availability

The data presented in this study are available on request from the corresponding author. The data are not publicly available due to confidentiality.

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
