# Peer review of "Identification of Transporter Polymorphisms Influencing Metformin Pharmacokinetics in Healthy Volunteers"

_jpm, 2023, doi:10.3390/jpm13030489_

Round 1
Reviewer 1 Report
jpm-2262396. Identification of transporter polymorphisms influencing metformin pharmacokinetics in healthy volunteers by Saiz-Rodríguez et al.
In this work, the authors explored the participation of 15 polymorphisms in 10 genes (members of the solute-carrier family and the ATP-binding cassette family), and their potential relationship with the pharmacokinetics of metformin in 176 healthy controls from Spain (to avoid confounding factors). They also evaluated the levels of the drug using HPLC-MS/MS.
I found the paper very interesting because metformin is widely used as an oral DM2 drug, and its pharmacokinetics and response exhibit a great variability among individuals.
I have the following comments:
In Methods, please, clarify
- Were all p-values ​​adjusted for multiple comparisons?
In the Results section:
Eighty-five subjects were self-identified as Caucasians, 89 were Latin and Mixed and 2 were Black.
-Did you perform ancestry of the samples?
-Following with the same issue, social scientists have long understood race to be a social construct, not an accurate representation of human genetic variation. Please, homogenize to one single term “race / ethnicity” along the manuscript, including Tables. Please, clarify the statement: “subjects were self-identified as Latin and Mixed.”
These terms must be well justified since an important variable to be considered in the pharmacokinetics of this drug in the study was: ethnicity.
-Regarding the “Safety profile” […] “In all, 20 volunteers (11.4%) experienced an ADR, being the most frequent headache (5.7%), followed by diarrhea (2.8%) and nausea (1.7%).” […]
-The three mentioned ADRs summed up 10.2%, did the authors miss one ADR? or the written numbers of the ADRs are wrong (a 1.2% is missing).
-Please, homogenize p-values with 2 decimal places in Table 2. Throughout the manuscript is written p<0.05
-The first paragraph of the Discussion section must be deleted because it comes from the Journal template: ”Authors should discuss the results and how […]”
-Authors should include limitations of its study. Such as sample size for ethnicities different from Caucasian, genotyping SNPs in only 10 genes, etc.
(e.g., “Our results align with these previous findings, since Black participants had a significantly greater AUC after adjusting for dosage and weight. However, […] just two of the participants were Black.”)
Author Response
Reviewer #1
In this work, the authors explored the participation of 15 polymorphisms in 10 genes (members of the solute-carrier family and the ATP-binding cassette family), and their potential relationship with the pharmacokinetics of metformin in 176 healthy controls from Spain (to avoid confounding factors). They also evaluated the levels of the drug using HPLC-MS/MS. I found the paper very interesting because metformin is widely used as an oral DM2 drug, and its pharmacokinetics and response exhibit a great variability among individuals.
Response:
We thank the reviewer for his kind words and are pleased that he/she found our manuscript to be of interest.
I have the following comments:
In Methods, please, clarify
- Were all p-values ​​adjusted for multiple comparisons?
Response: Thank you for your question. For better clarifying, we have included this paragraph in methods section: “As this study has an observational exploratory design, we did not adjust p values for multiple comparisons, according to which some authors recommend (22–24).”
Moreover, we have also added this sentence into the Study limitations section (page 11): “Our main limitation is the lack of multiple comparison correction, which could have led to false positive results. However, some experts recommend not to correct for multiple testing when analysing data [22–24]. Indeed, it is recommended to account for multiple comparison once interpreting the results, rather than in calculations. In fact, some authors state that the use of multiple comparison correction should be avoided to perform empirical research, since there is a potential cost of many more false negatives when controlling for false positives [41,42].”
In the Results section:
Eighty-five subjects were self-identified as Caucasians, 89 were Latin and Mixed and 2 were Black.
-Did you perform ancestry of the samples?
-Following with the same issue, social scientists have long understood race to be a social construct, not an accurate representation of human genetic variation. Please, homogenize to one single term “race / ethnicity” along the manuscript, including Tables. Please, clarify the statement: “subjects were self-identified as Latin and Mixed.”
These terms must be well justified since an important variable to be considered in the pharmacokinetics of this drug in the study was: ethnicity.
Response: We did not performed ancestry of the samples. Subjects were self-classified based on their personal origins. Moreover, we have homogenized the term race to ethnicity when appeared.
For a better understanding we have modified our results based on the PharmGKB recommendations which uses a system of nine biogeographical groups to annotate racial and ethnicity information about participants in pharmacogenomic studies. Seven of the nine groups are based on the geographical distribution of common genetic ancestry, which can be found here: https://www.pharmgkb.org/page/biogeographicalGroups (Huddart R, Fohner AE, Whirl-Carrillo M, Wojcik GL, Gignoux CR, Popejoy AB, Bustamante CD, Altman RB, Klein TE. Standardized Biogeographic Grouping System for Annotating Populations in Pharmacogenetic Research. Clin Pharmacol Ther. 2019 May;105(5):1256-1262. doi: 10.1002/cpt.1322.)
According to these, we have changed the groups into: European, Latino and African American. All these categories have been changed throughout the manuscript, including tables. This information has been included in methods section: “For ethnic groups comparisons, we have used the PharmGKB recommendations, which uses a system of nine biogeographical groups to annotate racial and ethnicity in-formation about participants in pharmacogenomic studies [20]. The groups are based on the geographical distribution of common genetic ancestry. According to these, we have identified and categorized the volunteers into three groups: European, Latino and African American.”
-Regarding the “Safety profile” […] “In all, 20 volunteers (11.4%) experienced an ADR, being the most frequent headache (5.7%), followed by diarrhea (2.8%) and nausea (1.7%).” […]
-The three mentioned ADRs summed up 10.2%, did the authors miss one ADR? or the written numbers of the ADRs are wrong (a 1.2% is missing).
Response: Thank you for your comment, we have added more information to clarify it: “In all, 20 subjects experienced an ADR, which is the 11.36% of the 176 total. 10 volunteers had headache (5.7%), 5 had diarrhea (2.8%) and 3 had nausea (1.7%), 1 each had hyploglycaemia, stomachache and dizziness (0.6% each).”
-Please, homogenize p-values with 2 decimal places in Table 2. Throughout the manuscript is written p<0.05
Response: thank you for your suggestion, p-values have been homogenized with 2 decimals in table 2.
-The first paragraph of the Discussion section must be deleted because it comes from the Journal template: ”Authors should discuss the results and how […]”
Response: thank you for noticing, it has been deleted.
-Authors should include limitations of its study. Such as sample size for ethnicities different from Caucasian, genotyping SNPs in only 10 genes, etc.
(e.g., “Our results align with these previous findings, since Black participants had a significantly greater AUC after adjusting for dosage and weight. However, […] just two of the participants were Black.”)
Response: thank you for your suggestion, a Study limitations section has been included at the end of discussion. “Our main limitation is the small proportion of African American in our population, which preclude us from finding a higher proportion of volunteers in this ethnic group. Moreover, as a candidate-gene study, there is a possibility that other variants could be related to metformin pharmacokinetics and were not included in the analysis. Another limitation is the lack of multiple comparison correction, which could have led to false positive results. However, some experts recommend not to correct for multiple testing when analyzing data [23–25]. Indeed, it is recommended to account for multiple comparison once interpreting the results, rather than in calculations. In fact, some authors state that the use of multiple comparison correction should be avoided to perform empirical research, since there is a potential cost of many more false negatives when controlling for false positives [42,43].“

Reviewer 2 Report
Saiz-Rodríguez and colleagues investigate the pharmacokinetics and pharmacogenetics of metformin. The subject matter is of importance.
Specific comments include:
* Please provide original references for some of the statements in the Introduction, not just reviews.
* Polymorphisms shouldd be checked and reported as per Higgins et al. 2021.
* The EUDRA-CT trials should be referenced fully.
* Please spell out CEGEN-PRB3-ISCIII. It may refer to a center in Spain that the rest of the world might not be not intimately familiar with.
* How were the polymorphisms tested chosen? Ditto perhaps for the genes?
* Is there a pressing need for tables 1 and 2 in the publication since the relevant data then shows up in table 3? Should tables 1 and 2 be relegated to supplementary materials?
* Delete the first paragraph of the Discussion........
Author Response
Reviewer #2:
Saiz-Rodríguez and colleagues investigate the pharmacokinetics and pharmacogenetics of metformin. The subject matter is of importance.
Specific comments include:
* Please provide original references for some of the statements in the Introduction, not just reviews.
Response: Thank you for your suggestion. References have been revised and two new references have been added. Most of them are original articles and clinical trials reports.
* Polymorphisms should be checked and reported as per Higgins et al. 2021.
Response: Thank you for your suggestion, however, we were unable to find the reference. Please provide the DOI or PMID.
* The EUDRA-CT trials should be referenced fully.
Response: Thank you for your suggestion. However, we have mentioned in the methods section the full reference of the trials, which are: EUDRA-CT: 2019-001393-29; EUDRA-CT: 2017-005145-79; EUDRA-CT: 2018-000401-23; EUDRA-CT: 2017-004727-73; EUDRA-CT: 2019-003274-79; EUDRA-CT: 2020-003049-12; EUDRA-CT: 2020-003619-81; EUDRA-CT: 2020-004728-40.
* Please spell out CEGEN-PRB3-ISCIII. It may refer to a center in Spain that the rest of the world might not be not intimately familiar with.
Response: Thank you for your suggestion. It has been corrected.
* How were the polymorphisms tested chosen? Ditto perhaps for the genes?
Response: Thank you for your question. The genes and polymorphisms were chosen for their implication in metformin transport, and the functional consequences of the polymorphisms, based on previous literature and minor allele frequencies. This information has been added to the manuscript for greater clarity.
* Is there a pressing need for tables 1 and 2 in the publication since the relevant data then shows up in table 3? Should tables 1 and 2 be relegated to supplementary materials?
Response: Thank you for your suggestion. However, as no other reviewer has suggested a restructuring of the tables to supplementary material, we consider it a good option to leave them in the main text. However, if the reviewers and/or editors consider it, they would be modified to be supplementary material.
* Delete the first paragraph of the Discussion........
Response: Thank you for your suggestion. It has been deleted.

Reviewer 3 Report
In this article, the authors provide the comprehensive analysis of individual differences in the pharmacological response to metformin. The assessment of a set of the independent parameters (age, sex, race, polymorphism of the ABCB1, SLC2A2, SLC22A2, or SLC47A1 genes) to pharmacokinetic of the drug was performed. However, I would like to mention the several comments:
1. The authors repeatedly mention in the manuscript about the analysis of the ethnicity to the therapeutic response, however, in fact, they assess the race but not ethnicity.
2. I suppose that if the authors presented the information about the study group in the form of a table, it would greatly improve the perception of the article. Also I would like to get more information about donors and their health status.
3. In my opinion, it is unreasonable to draw any conclusions about the influence of the race on the pharmacological response based on a such a small group (2 Black individuals).
4. How do you explain the lack of Hardy-Weinberg equilibrium for ABCG2 (rs2231142) and SLC22A1 (rs72552763, rs12208357 and rs34059508)?
5. In the Discussion section, apparently, the phrase «Authors should discuss the results and how they can be interpreted from the perspective of previous studies and of the working hypotheses. The findings and their implications should be discussed in the broadest context possible. Future research directions may also be highlighted» was inserted by mistake.
Author Response
Reviewer #3:
In this article, the authors provide the comprehensive analysis of individual differences in the pharmacological response to metformin. The assessment of a set of the independent parameters (age, sex, race, polymorphism of the ABCB1, SLC2A2, SLC22A2, or SLC47A1 genes) to pharmacokinetic of the drug was performed. However, I would like to mention the several comments:
- The authors repeatedly mention in the manuscript about the analysis of the ethnicity to the therapeutic response, however, in fact, they assess the race but not ethnicity.
Response: Thank you for your comment. For a better understanding we have modified our results based on the PharmGKB recommendations which uses a system of nine biogeographical groups to annotate racial and ethnicity information about participants in pharmacogenomic studies. Seven of the nine groups are based on the geographical distribution of common genetic ancestry, which can be found here: https://www.pharmgkb.org/page/biogeographicalGroups
According to these, we have changed the groups into: European, Latino and African American. All these categories have been changed throughout the manuscript, including tables.
Moreover, according to the comments raised by reviewer #1 we have homogenised the terminology to ethnicity throughout the manuscript.
- I suppose that if the authors presented the information about the study group in the form of a table, it would greatly improve the perception of the article. Also, I would like to get more information about donors and their health status.
Response: Thank you for your suggestion. However, we believe that the manuscript already has a large number of tables containing the genetic information and its implication on pharmacokinetic parameters. The information regarding sex, age and weight is correctly reflected at the beginning of the results section.
Regarding the volunteer’s health status, it is described in the methods section “The inclusion criteria were: male and female volunteers aged from 18 to 55, free from organic or psychiatric conditions. The exclusion criteria were: history of kidney and/or liver damage, drug intake 48 h before receiving the study medication, having body mass index outside the 18.5–30.0 kg/m2 range, history of sensitivity to any drug and positive drug screening, smoker and daily alcohol consumer, blood donation and pregnant or breastfeeding women.” Therefore, there is no condition or concomitant medication to consider.
- In my opinion, it is unreasonable to draw any conclusions about the influence of the race on the pharmacological response based on a such a small group (2 Black individuals).
Response: Thank you for your suggestion. We agree with the reviewer that no solid conclusions can be drawn with a small sample size, so we have reflected this fact in the study limitations section. Nevertheless, it is necessary to highlight this tendency so that it can be taken into account in future approaches as another variable to be included.
- How do you explain the lack of Hardy-Weinberg equilibrium for ABCG2 (rs2231142) and SLC22A1 (rs72552763, rs12208357 and rs34059508)?
Response: Thank you for your question. It might be attributable to the small sample size, but we consider that it does not influence the evaluation of the association analysis with pharmacokinetic parameters.
- In the Discussion section, apparently, the phrase «Authors should discuss the results and how they can be interpreted from the perspective of previous studies and of the working hypotheses. The findings and their implications should be discussed in the broadest context possible. Future research directions may also be highlighted» was inserted by mistake.
Response: Thank you for your suggestion. It has been deleted.

Round 2
Reviewer 1 Report
The authors have responded to all my queries. The manuscript can be accepted in its present form.
Reviewer 2 Report
The authors have made some changes to address some of the issues raised by this referee.
Reviewer 3 Report
Thanks for the clarification and the adjustments made to the paper.